

# Effective fractonic behavior
# in a two-dimensional exactly solvable spin liquid

Guilherme Delfino[1*], Weslei Bernardino Fontana[2†],
Pedro R. S. Gomes[3‡] and Claudio Chamon[1∘]

**1** Physics Department, Boston University, Boston, MA, 02215, USA
**2** International Institute of Physics, Universidade Federal
do Rio Grande do Norte, 59078-970 Natal-RN, Brasil
**3** Departamento de Física, Universidade Estadual de Londrina,
86057-970, Londrina, PR, Brasil

⋆ delfino@bu.edu , † weslei@uel.br , ‡ pedrogomes@uel.br , ∘ chamon@bu.edu

## Abstract

In this work we propose a $\mathbb{Z}_N$ clock model which is exactly solvable on the lattice. We find exotic properties for the low-energy physics, such as UV/IR mixing and excitations with restricted mobility, that resemble fractonic physics from higher dimensional models. We then study the continuum descriptions for the lattice system in two distinct regimes and find two qualitative distinct field theories for each one of them. A characteristic time scale that grows exponentially fast with $N^2$ (and diverges rapidly as function of system parameters) separates these two regimes. For times below this scale, the system is described by an effective fractonic Chern-Simons-like action, where higher-form symmetries prevent quasiparticles from hopping. In this regime, the system behaves effectively as a fracton as isolated particles, in practice, never leave their original position. Beyond the large characteristic time scale, the excitations are mobile and the effective field theory is given by a pure mutual Chern-Simons action. In this regime, the UV/IR properties of the system is captured by a peculiar realization of the translation group.

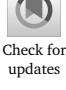
# 1 Introduction

Topological order [1] – after few decades of progress and developments – has become an established paradigm for describing a rich landscape of quantum states of matter. Among states recognized to be topologically ordered, the quintessential representatives are the fractional quantum Hall phases [2] and quantum spin liquids [3]. Telltale features of topological order are: (i) a ground state degeneracy that depends on the topology (genus) of the manifold on which the system is placed; and (ii) the presence of quasiparticles with fractionalized charges and statistics. Solvable lattice models have largely helped the understanding of topological order, with Kitaev's toric code [4] and Wen's plaquette model [5] as prime examples of simple Hamiltonians that downright capture the essence of the underlying physics. In the continuum, the low-energy physics of two-dimensional topological order is elegantly captured in terms of 2+1D Chern-Simons field theories [6,7].

Fracton systems [8–29] are novel 3D topological phases with features that depart from those in canonical topological order: the quasiparticle excitations – fractons – either show restricted mobility (type I) or are immobile altogether (type II). The mobility restrictions follow in general from the presence of generalized symmetries known as subsystem symmetries [13, 24, 30, 31]. In addition, some geometric data like the lattice spacing is also needed to determine the ground state degeneracy. This implies a kind of UV/IR mixing in fractonic topological order.

Recently, certain lattice models have been obtained by Higgsing a rank-2 $U(1)$ lattice gauge theory, yielding to $\mathbb{Z}_N$ quantum spin liquids [32–37]. These models display features resembling those of fracton systems, but the elementary excitations, in contrast to fractons, can still hop, not in lattice spacing steps but in steps whose size scales linearly with $N$. In such systems, there exists an explicit interplay between the topological order properties and the translation symmetry, where different anyons are allowed to map into each other when acted by a translation. These systems are said to have symmetry enriched topological order [38–40].

In this paper, instead of Higgsing a lattice gauge theory, we construct a 2D lattice model with similar properties by collapsing onto a plane the $\mathbb{Z}_N$ version of the 3D type I fracton model of Ref. [8]. The resulting 2D model exhibits the following features: (i) the Hamiltonian is a sum of commuting projectors (hence exactly solvable); (ii) the ground state degeneracy depends on both $N$ and the geometry of the lattice; (iii) both charge and dipole moment of the excitations are conserved quantities, resulting in mobility restrictions; (iv) there are nontrivial mutual statistics among the emergent quasiparticles; and (v) the isolated excitations can hop only by integer steps of size $N$.

In both the Higgsed gauge models and the flattened model, when the system is placed on a torus, the ground state degeneracy, like those in fractons, also depend on the commensuration (via the greatest common divisor) of the system size dimensions $L_x$ and $L_y$ with $N$.

These "quasi-fracton" [32–37] systems thus bridge canonical topological order with fracton topological order. Here we explore this bridge and argue that, as function of $N$ and the coupling constants in the model, an exponentially large window of time opens in which the system behaves effectively as a fracton in $d = 2$.

The basic idea is that, while an isolated excitation can hop in a step size of order $N$, the tunneling rate for this process (as we show in the paper) scales exponentially with $N^2$. The characteristic time for an isolated excitations to move is $\tau_{\text{monopole}} \sim (J/g)^{N(N+1)/2}$, where $J$ is the energy scale in our exactly solvable model, and $g$ is the scale of coupling that gives mobility to the excitations (either the scale of a transverse field or the coupling to a Caldeira-Leggett dissipative bath). This time scale is to be contrasted to that for dipolar motion, $\tau_{\text{dipole}} \sim (J/g)$, which is independent of $N$. Effectively, this separation of time scales means that while dipolar motion is always present for modestly large ratios $J/g$, an isolated excitation may take times larger than the age of the universe (even for quite conservatively small $N$) to move. In this sense, a theory with such properties behaves effectively as a two-dimensional fracton system.

In this paper we explore this separation of time scales and study the effective field theory description of the lattice model in two regimes. In the scale of times $\tau \ll \tau_{\text{monopole}}$, in which the quasiparticles do not move, we obtain a fractonic Chern-Simons effective field theory description, i.e., a Chern-Simons-like action with higher order derivatives similarly to Ref. [16, 41, 42]. The field theory is constructed via a bosonization of sorts [43], similarly to that used in Ref. [44] for the same 3D fracton that, upon flattening to 2D, yields the lattice model here studied. Our effective theory is an alternative to the Higgsed rank-2 $U(1)$ theories of Refs. [32–37]. The Chern-Simons-like theory captures many of the features of the lattice model, such as dipole conservation, the fact that the system is gapped, and the mutual statistics of quasi-fractons and quadrupoles.

For times longer than the (exponentially large) scale $\tau_{\text{monopole}}$, the higher multipole moment conservation fails to hold. The intuition from the lattice model is that the anyons can freely move along the entire system if one waits long enough. The only universal properties that survive this infinite time limit is the topological mutual statics among the quasiparticles. We use such data to write down a $[U(1)]^4$ continuum mutual Chern-Simons theory and use this effective description to explicitly compute the ground state degeneracy. In this regime, the UV/IR aspects of the theory are implemented through the boundary conditions on the gauge fields on a compact space [37].

The paper is organized as follows. In Sec. 2 we introduce the lattice model and discuss its properties, as the ground state degeneracy, and the low-energy excitations. In Sec. 3 we discuss the effective continuum description of the model in the two time-scale regimes, and study their properties. We conclude in Sec. 4.

# 2 Lattice Model

The model we study here corresponds to the collapse of the $\mathbb{Z}_N$ octahedron operators of the Chamon code [8, 9] onto a plane. As we shall see, the resulting two-dimensional model turns out to be a dipole conserving spin liquid, where quadrupole bound states are free to move throughout the system, dipole bound states are lineons, and there are single excitations that can only hop in steps of size $N$ units of the lattice spacing.

## 2.1 The Model

We start by considering $\mathbb{Z}_N$ degrees of freedom associated with each site of a two-dimensional square lattice. The operators that act at each site are the generalized $\mathbb{Z}_N$ "clock" $Z$ and "shift" $X$ Pauli operators. They can be represented in terms of unitary and traceless $N \times N$ matrices

that realize the $\mathbb{Z}_N$ algebra

$$X_{\vec{x}} Z_{\vec{y}} = \omega^{\delta_{\vec{x},\vec{y}}} Z_{\vec{y}} X_{\vec{x}}, \quad \text{with} \quad \omega \equiv e^{\frac{2\pi i}{N}}, \tag{1}$$

where $\vec{x}, \vec{y}$ label the lattice sites. Both $Z$ and $X$ obey $Z^N = X^N = \mathbb{1}$ and, as unitary matrices, have complex eigenvalues $1, \omega, \ldots, \omega^{N-1}$. For $N = 2$, they reduce to the usual Hermitian Pauli matrices.

Let us consider the two-dimensional lattice spanned by the vectors $\vec{a}_1$ and $\vec{a}_2$, with a $\mathbb{Z}_N$ degree of freedom on every site $\vec{x} = (\hat{x}, \hat{y}) \equiv \hat{x}\vec{a}_1 + \hat{y}\vec{a}_2$, as shown in Fig. 1. We define the

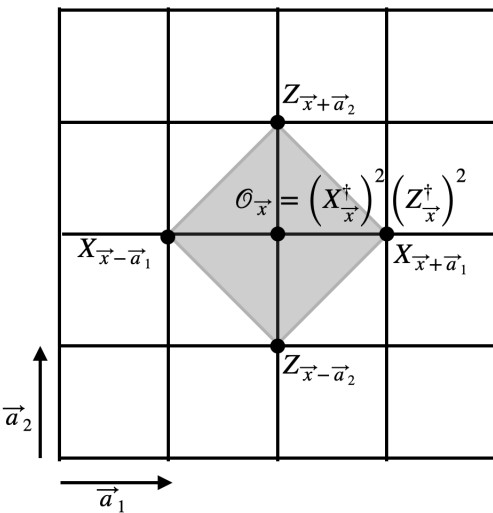

Figure 1: Lattice spanned by the lattice vectors $\vec{a}_1$ and $\vec{a}_2$ and the plaquette operators $B_{\vec{x}}$.

lattice Hamiltonian to be a sum of site-centered plaquette operators

$$H = -\frac{J}{2} \sum_{\vec{x}} \left( B_{\vec{x}} + B_{\vec{x}}^{\dagger} \right), \tag{2}$$

where the plaquette operator is

$$B_{\vec{x}} \equiv X_{\vec{x}-\vec{a}_1} Z_{\vec{x}-\vec{a}_2} \mathcal{O}_{\vec{x}} X_{\vec{x}+\vec{a}_1} Z_{\vec{x}+\vec{a}_2}, \tag{3}$$

with

$$\mathcal{O}_{\vec{x}} \equiv \left( X_{\vec{x}}^{\dagger} \right)^2 \left( Z_{\vec{x}}^{\dagger} \right)^2, \tag{4}$$

as illustrated in Fig. 1. The $\mathcal{O}_{\vec{x}}$-term is introduced to keep the $B_{\vec{x}}$ plaquette operators neutral under the $\mathbb{Z}_N$ group. This model can be interpreted as a two-dimensional, squeezed version of the $\mathbb{Z}_N$ Chamon code [8], in the sense that the octahedron operators are collapsed onto a plane. As the octahedron operators are squished into the $xz$ plane, the two operators $Y^{\dagger} = (iXZ)^{\dagger}$ are taken to the center of the two-dimensional plaquette and gives rise to the $\mathcal{O}_{\vec{x}} \sim (Y^{\dagger})^2$ operator, as shown in Figure 2. For the $N = 2$ case, the model reduces to two copies of the $\mathbb{Z}_2$ Wen plaquette model [5].

This model is exactly solvable since the Hamiltonian is given in terms of commuting projectors, i.e., all the terms in the Hamiltonian (2) are simultaneously commuting,

$$\left[ B_{\vec{x}}, B_{\vec{y}} \right] = \left[ B_{\vec{x}}, B_{\vec{y}}^{\dagger} \right] = \left[ B_{\vec{x}}^{\dagger}, B_{\vec{y}}^{\dagger} \right] = 0. \tag{5}$$

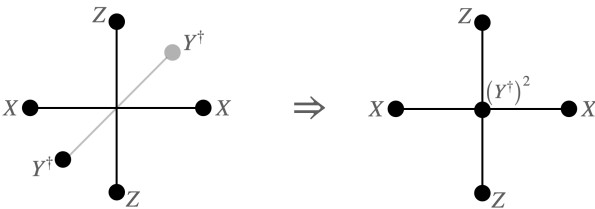

Figure 2: The octahedron operator in the Chamon code (in the left) is squeezed into the $xz$ plane (in the right).

These commutation relations follow from all the possible ways that two distinct plaquette operators can share common sites, as depicted in Fig.(3). Then, using the $\mathbb{Z}_N$ algebra, it is simple to show that (5) holds for any two sites $\vec{x}$ and $\vec{y}$ of the lattice. In the following, we study its physical properties.

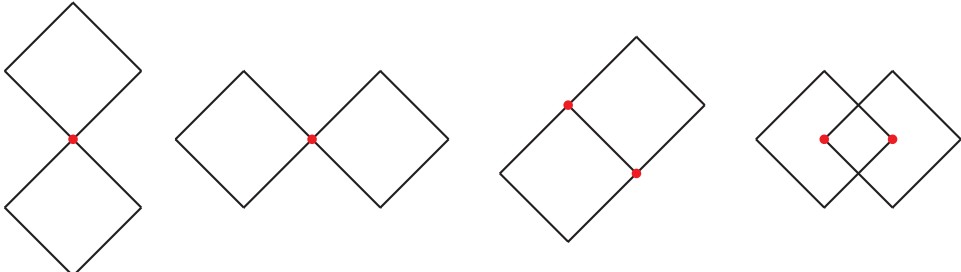

Figure 3: Possibilities for two plaquette operators to share lattice sites (in red).

## 2.2 Ground State Degeneracy

The plaquette operators obey $B_{\vec{x}}^N = \mathbb{1}$ for every point $\vec{x}$ in the lattice. Therefore, in the same way as the $\mathbb{Z}_N$ operators, they have $N$ eigenvalues: the $N$ roots of identity in a unit circle $1, \omega, \omega^2, \dots, \omega^{N-1}$. The ground state space $\mathcal{H}_0$ contains the states that maximize the real part of the eigenvalues of the plaquette operators $B_{\vec{x}} \to 1$,

$$\mathcal{H}_0 = \{|\psi_0\rangle : B_{\vec{x}}|0\rangle = |0\rangle \text{ for all } \vec{x} \text{ in the lattice}\} . \tag{6}$$

The states $|\psi_0\rangle \in \mathcal{H}_0$ are topologically ordered since, as we shall see, they are gapped and sensitive to the lattice topology. States above the ground state are states with a violated plaquette $\frac{1}{2}(B_{\vec{x}} + B_{\vec{x}}^\dagger)|\psi\rangle = \cos(2\pi/N)|\psi\rangle$ for some site $\vec{x}$. For any finite $N$, there is a gap between the ground state $|\psi_0\rangle$ and the first excited states

$$\Delta E = J\left(1 - \cos\left(\frac{2\pi}{N}\right)\right). \tag{7}$$

In a periodic lattice, the ground state degeneracy is nontrivial, that is, $\dim \mathcal{H}_0 > 1$. Note that for the lattice with the number of sites $\mathsf{N}_{\text{sites}}$, the dimension of the total Hilbert space $\mathcal{H}$ is $N^{\mathsf{N}_{\text{sites}}}$ but there are not as many $B_{\vec{x}}$ eigenvalues to label all these states. Due to global constraints, not all the $N$ eigenvalues of the $\mathsf{N}_{\text{sites}}$ operators $B_{\vec{x}}$ are independent: the product of all the plaquette operators of the lattice satisfies

$$\prod_{\vec{x}} B_{\vec{x}} = \mathbb{1}. \tag{8}$$

This constraint implies that all the states are at least $N$-fold degenerate.

There is, however, the possibility of additional global constraints depending on the relation between $N$ and the linear sizes $L_x$ and $L_y$ of the lattice. In fact, if $N$ and $L_x$ are not co-primes, i.e, their greatest common divisor is $\gcd(N, L_x) \neq 1$, it follows that

$$\prod_{\vec{x}=(x,y)} (B_{\vec{x}})^{\hat{x}\rho_x} = \mathbb{1}, \quad \text{with} \quad \rho_x \equiv \frac{N}{\gcd(N, L_x)}, \tag{9}$$

is also a global constraint of the system. The same holds for the vertical direction if $\gcd(N, L_y) \neq 1$,

$$\prod_{\vec{x}=(x,y)} (B_{\vec{x}})^{\hat{y}\rho_y} = \mathbb{1}, \quad \text{with} \quad \rho_y \equiv \frac{N}{\gcd(N, L_y)}. \tag{10}$$

Finally, depending on $N, L_x$ and $L_y$ altogether, we have the further constraint

$$\prod_{\vec{x}=(x,y)} (B_{\vec{x}})^{\hat{x}\hat{y}\rho_{xy}} = \mathbb{1}, \quad \text{with} \quad \rho_{xy} \equiv \frac{N}{\gcd(N, L_x, L_y)}. \tag{11}$$

For every independent global constraint, the states have their degeneracy increased

$$\dim \mathcal{H}_0 = N \gcd(N, L_x) \gcd(N, L_y) \gcd(N, L_x, L_y), \tag{12}$$

which lies in the interval $N \leq \dim \mathcal{H}_0 \leq N^4$. Notice that it depends explicitly on the interplay between the group order and the lattice size, which is a typical property of fractonic systems [45]. In the case $N = 2$, the model reduces to the usual Wen plaquette model [5]. More precisely, for $L_x$ and $L_y$ even, the model reduces to two copies of the Wen plaquette model, corresponding to a $\mathbb{Z}_2 \times \mathbb{Z}_2$ topological order, with $\dim \mathcal{H}_0 = 2^4$ and the four global constraints on $B_{\vec{x}}$ being just the odd and even sub-lattice constraints [5]. For $L_x, L_y$ and $N = 2$ co-primes, the system reduces to a single Wen plaquette model with linear dimensions $2L_x$ and $2L_y$ and $\dim \mathcal{H}_0 = 2$. For the case in which $L_x = L_y = 0 \mod N$, the topological ground state degeneracy $\dim \mathcal{H}_0 = N^4$ suggests that the model realizes $\mathbb{Z}_N \times \mathbb{Z}_N$ topological order. Furthermore, in section 3.2, we show that for arbitrary $L_x, L_y$ and $N$ the low-energy physics of the model is incorporated into a double $\mathbb{Z}_N$ BF effective field theory, again indicating $\mathbb{Z}_N \times \mathbb{Z}_N$ topological order.

## 2.3 Excitations with Restricted Mobility

Excitations above the ground state are localized in space and correspond to states with at least one of the eigenvalues of the $B_{\vec{x}}$ operators different from 1. Let us consider states with general eigenvalues $B_{\vec{x}} |\psi\rangle = e^{\frac{2\pi i q_{\vec{x}}}{N}} |\psi\rangle$, where we interpret $q_{\vec{x}}$ as the $\mathbb{Z}_N$ charge defined mod $N$, associated with the excitation localized at the position $\vec{x}$. In terms of the charge $q_{\vec{x}}$, the global constraints (8), (9), (10), and (11) translate into conservation of charge, $x$ and $y$ dipole moments, and the off-diagonal quadrupole moment,

$$
\begin{aligned}
Q &= \sum_{\vec{x}} q_{\vec{x}} = 0 \mod N, \\
P_x &= \sum_{\vec{x}} \hat{x} \, q_{\vec{x}} = 0 \mod \gcd(N, L_x), \\
P_y &= \sum_{\vec{x}} \hat{y} \, q_{\vec{x}} = 0 \mod \gcd(N, L_y), \\
Q_{xy} &= \sum_{\vec{x}} \hat{x} \, \hat{y} \, q_{\vec{x}} = 0 \mod \gcd(N, L_x, L_y),
\end{aligned}
\tag{13}
$$

respectively. These conservation laws impose restrictions in the way the excitations can propagate in the system from one position to another. As we shall discuss in the following, given a specific type of excitation in a particular lattice position $\vec{x}$, there are regions in the lattice that are inaccessible for such excitations.

Let us consider an isolated excitation with unit charge $B_{\vec{x}_0}|\psi\rangle = e^{\frac{2\pi i}{N}}|\psi\rangle$ located at $\vec{x}_0 = (\hat{x}_0, \hat{y}_0)$. Its hopping is restricted to happen in steps of $N$ lattice units either in the $x$ or $y$ directions. To understand this point, we consider a rigid string $\lambda$ of length $N$ with initial and final points at $\vec{x}_0$ and $\vec{x}_f$. Then, we can define the operators supported on such string,

$$
U(\lambda) \equiv
\begin{cases}
\displaystyle\prod_{\hat{x}=\hat{x}_0}^{\hat{x}_0+N} (Z_{\vec{x}})^{\hat{x}}, & \text{if } \lambda \text{ is horizontally oriented } \lambda = \lambda_x\,, \\
\displaystyle\prod_{\hat{y}=\hat{y}_0}^{\hat{y}_0+N} (X_{\vec{x}})^{\hat{y}}, & \text{if } \lambda \text{ is vertically oriented } \lambda = \lambda_y\,,
\end{cases}
\tag{14}
$$

that are able to hop these excitations to the final positions $\vec{x}_f = \vec{x}_0 + N\vec{a}_1$ and $\vec{x}_f = \vec{x}_0 + N\vec{a}_2$, respectively. The existence of such hopping operators follow from the conservation of $Q$ mod $N$ in the first line of (13). Its effect on the ground state is to create an excitation of charge 1 at $\vec{x}_0$ and $-1$ (mod $N$) at $\vec{x}_f$. The class of $N$-step hopping line operators (14) can also be found in Higgsed phases of symmetric tensor gauge theories [32, 33, 35].

The action of $U$ on the ground state can be understood from its commutation properties with the plaquette operators $B_{\vec{x}}$. Along the string $\lambda$, $[U(\lambda), B_{\vec{x}}] = 0$ and no excitations are created. In contrast, at its endpoints $\vec{x}_0$ and $\vec{x}_f$,

$$
U(\lambda)B_{\vec{x}_0} = \omega B_{\vec{x}_0} U(\lambda) \quad \text{and} \quad U(\lambda)B_{\vec{x}_f} = \omega^{N-1} B_{\vec{x}_f} U(\lambda)\,,
\tag{15}
$$

creating quasiparticles of charges $q_{\vec{x}_0} = 1$ and $q_{\vec{x}_f} = -1$, which we refer to as q-particles. For a system with periodic boundary conditions, depending on the relation between $N$, $L_x$ and $L_y$, we need to hop the q-particles more than once around the system through applications of $U$ to be able to return to their original position. For the general case, the translation operations

$$
\hat{T} : (\hat{x}, \hat{y}) \mapsto \left(\hat{x} + \text{lcm}(N, L_x), \hat{y} + \text{lcm}(N, L_y)\right),
\tag{16}
$$

allow the $U$ operators to close on themselves, where lcm stands for the least common multiple. The action of $\hat{T}$ on the q−anyons will be useful in section 3.2, where we find an effective field theory for this lattice model. While the action of arbitrary translations is rather complicated, the operations $\hat{T}$ act as identities on the anyon space and will be enough to recover the ground state degeneracy in the deep IR description. As we will argue, the commensurability of $N$ with the linear system sizes $L_x$ and $L_y$ plays a role in the low-energy properties of the system, reminiscent of fractonic physics.

Composed excitations also present restricted mobility. Indeed, dipole configurations emerge as quasiparticles in the excited states and correspond to excitations that can move only along rigid lines, i.e., they behave precisely as the lineons of fracton systems [46]. Let $\gamma_x$ and $\gamma_y$ be vertical or horizontal oriented straight lines of arbitrary length. We define the line operators

$$
V(\gamma) \equiv
\begin{cases}
\prod_{\gamma} Z_{\vec{x}}, & \text{if } \gamma \text{ is horizontally oriented}\,, \\
\prod_{\gamma} X_{\vec{x}}, & \text{if } \gamma \text{ is vertically oriented}\,,
\end{cases}
\tag{17}
$$

which are responsible to create the charge distributions shown in Fig.(4) when acting on the vacuum. These operators create dipoles that are oriented in the same direction as the corresponding string $\gamma$. We refer to these dipoles as $\mathfrak{p}^x$ and $\mathfrak{p}^y$-particles. Again, the rigidity of the

strings $\gamma$ imply that these excitations can move continuously only along their own axis. Thus, under translations around the system the $\mathfrak{p}^x$ and $\mathfrak{p}^y$-particles need to be translated according to

$$\hat{T} : (\hat{x}, \hat{y}) \mapsto (\hat{x} + L_x, \hat{y} + \mathrm{lcm}(N, L_y)) \quad \text{and} \quad \hat{T} : (\hat{x}, \hat{y}) \mapsto (\hat{x} + \mathrm{lcm}(N, L_y), \hat{y} + L_y), \quad (18)$$

respectively.

To conclude the discussion of these one-dimensional particles, it is useful to introduce two quasiparticles $\mathfrak{d}^1$ and $\mathfrak{d}^2$, created at the endpoints of the rigid line operators

$$S(\Gamma) \equiv \begin{cases} \prod_\Gamma \left( X_{\vec{x}} Z_{\vec{x}} \right), & \text{if } \Gamma \text{ is oriented along } \vec{a}_1 + \vec{a}_2, \\ \prod_\Gamma \left( X_{\vec{x}}^\dagger Z_{\vec{x}} \right), & \text{if } \Gamma \text{ is oriented along } \vec{a}_1 - \vec{a}_2, \end{cases} \quad (19)$$

for $\Gamma$ a rigid string, as indicated in Fig 4. The $\mathfrak{d}$ lineons are not independent from the $\mathfrak{p}$ ones, as the $S$ strings are built out of products of $V(\gamma_x)$ and $V(\gamma_y)$. In fact, these anyons are related by fusion, $\mathfrak{d}^1 = \mathfrak{p}^x \times \mathfrak{p}^y$ and $\mathfrak{d}^2 = \mathfrak{p}^x \times \overline{\mathfrak{p}^y}$, so that any pair among them contains enough information to fix the properties of the remaining one.

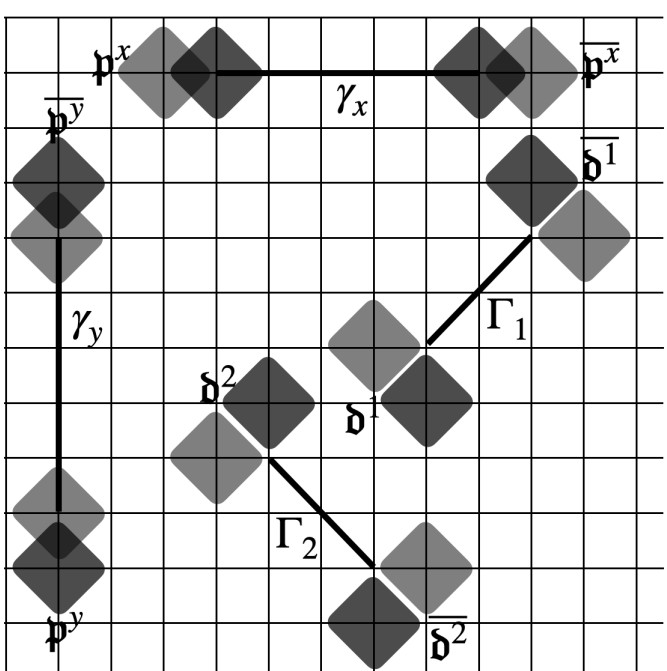

Figure 4: String operators $V(\gamma)$ horizontally and vertically oriented and also $S(\Gamma)$, diagonally oriented. Darker diamonds represent positive charges while lighter ones represent negative charges.

Finally, we have composite excitations that are completely mobile, created at the endpoints of the double-string operators

$$W_x \equiv \prod_{\vec{x} \in \gamma_x} Z_{\vec{x}} Z_{\vec{x}+\vec{a}_2}^\dagger \quad \text{and} \quad W_y \equiv \prod_{\vec{x} \in \gamma_y} X_{\vec{x}} X_{\vec{x}+\vec{a}_1}^\dagger . \quad (20)$$

We refer to these excitations as $\mathfrak{m}$-particles. Under translations

$$\hat{T} : (\hat{x}, \hat{y}) \mapsto (\hat{x} + L_x, \hat{y} + L_y), \quad (21)$$

the $\mathfrak{m}$-particles return to their original position.

## 2.4 Anyonic Mutual Statistics

Although $\mathfrak{q}$, $\mathfrak{p}^x$, $\mathfrak{p}^y$, and $\mathfrak{m}$-particles are all bosonic excitations, they can present nontrivial mutual statistics among themselves - a characteristic signature in quantum spin liquids. Since $\mathfrak{m}$ excitations are completely mobile, their mutual statistics with $\mathfrak{q}$, $\mathfrak{p}^x$, and $\mathfrak{p}^y$ (as well as $\mathfrak{d}^1$ and $\mathfrak{d}^2$) are the easiest ones to see. Moving a $\mathfrak{m}$-particle with charge $q_0'$ around a closed loop $C$, in a given state $|\psi\rangle$, corresponds to the application of the operator $\tilde{M}_C \equiv \left( W_y W_{x+L} W_{y+L}^\dagger W_x^\dagger \right)^{q_0'}$ on the state $|\psi\rangle$. A special feature about this operator is that it is the product of all plaquette operators inside $C$,

$$\tilde{M}_C |\psi\rangle = \prod_{\vec{x} \text{ inside } C} (B_{\vec{x}})^{q_0'} |\psi\rangle . \tag{22}$$

Thus, if $|\psi\rangle$ is a state containing, besides $\mathfrak{m}$, an isolated $\mathfrak{q}$-particle with charge $q_0$ located at $\vec{x}$, inside $C$, the state acquires a phase

$$\tilde{M}_C |\psi\rangle = \exp\left( \frac{2\pi i q_0' q_0}{N} \right) |\psi\rangle , \tag{23}$$

implying a nontrivial mutual statistics $\theta(\mathfrak{q},\mathfrak{m}) = 2\pi/N$. On the other hand, if $|\psi\rangle$ contains any charge configuration with a neutral $\mathbb{Z}_N$ charge, the mutual statistics of $\mathfrak{m}$ with such particles is trivial. Therefore, we conclude that the mutual statistics among $\mathfrak{m}$ and either $\mathfrak{p}^x$, $\mathfrak{p}^y$, $\mathfrak{d}^1$ or $\mathfrak{d}^2$ is trivial.

The mutual statistics among $\mathfrak{q}$, $\mathfrak{p}$ and $\mathfrak{d}$ can be seen from the algebra among the corresponding string operators $U$'s, $V$'s and $S$'s. The non-commuting algebra

$$V(\gamma_y)V(\gamma_x) = \omega V(\gamma_x)V(\gamma_y), \qquad\qquad S(\Gamma_2)S(\Gamma_1) = \omega^{-2}S(\Gamma_1)S(\Gamma_2),$$
$$S(\Gamma_1)V(\gamma_x) = \omega V(\gamma_x)S(\Gamma_1), \qquad\qquad S(\Gamma_2)V(\gamma_x) = \omega^{-1}V(\gamma_x)S(\Gamma_2),$$
$$V(\gamma_y)U(\lambda_x) = \omega^{\hat{x}} V(\gamma_y)U(\lambda_x), \qquad\qquad V(\gamma_x)U(\lambda_y) = \omega^{-\hat{y}} V(\gamma_x)U(\lambda_y),$$

imply that the corresponding pairs of particles share nontrivial mutual statistics. In these equations, $\hat{x}$ and $\hat{y}$ are the relative coordinates of the point where the two perpendicular strings $\lambda$ and $\gamma$ intersect, with respect to the starting point of $\lambda$. As an explicit example, consider the braiding among the $\mathfrak{p}^x$ and $\mathfrak{p}^y$-particles in the state $|\psi\rangle$, according to the process illustrated in Fig. 5. Using the commutation of the first relation in (24), it follows that

$$V^\dagger(\gamma_y)V^\dagger(\gamma_x)V(\gamma_y)V(\gamma_x) |\psi\rangle = \omega |\psi\rangle , \tag{24}$$

which implies that $\mathfrak{p}^x$ and $\mathfrak{p}^y$ have mutual statistics $\theta(\mathfrak{p}^x,\mathfrak{p}^y) = 2\pi/N \mod N$. Similarly, for the diagonal lineons, $\theta(\mathfrak{d}^1,\mathfrak{d}^2) = -4\pi/N$. Also, from the second line of equations in (24) and the fusion rules, $\mathfrak{d}^1 = \mathfrak{p}^x \times \mathfrak{p}^y$ and $\mathfrak{d}^2 = \mathfrak{p}^x \times \overline{\mathfrak{p}^y}$,

$$\theta(\mathfrak{p}^x,\mathfrak{d}^1) = \frac{2\pi}{N}, \qquad \theta(\mathfrak{p}^x,\mathfrak{d}^2) = -\frac{2\pi}{N},$$
$$\theta(\mathfrak{p}^y,\mathfrak{d}^1) = -\frac{2\pi}{N}, \qquad \theta(\mathfrak{p}^y,\mathfrak{d}^2) = -\frac{2\pi}{N}. \tag{25}$$

Finally, for the $\mathfrak{q}$-particles, the relations in the second line of (24) imply that their mutual statistics with $\mathfrak{p}^x$ and $\mathfrak{p}^y$ depend on their relative initial position $\mod N$,

$$V^\dagger(\gamma_x)U^\dagger(\gamma_y)V(\gamma_x)U(\gamma_y) |\psi\rangle = \omega^{-\hat{y} \mod N} |\psi\rangle \Rightarrow \theta(\mathfrak{q},\mathfrak{p}^x) = -\frac{2\pi\hat{y}}{N}, \tag{26}$$

$$V^\dagger(\gamma_y)U^\dagger(\gamma_x)V(\gamma_y)U(\gamma_x) |\psi\rangle = \omega^{\hat{x} \mod N} |\psi\rangle \Rightarrow \theta(\mathfrak{q},\mathfrak{p}^y) = \frac{2\pi\hat{x}}{N}, \tag{27}$$

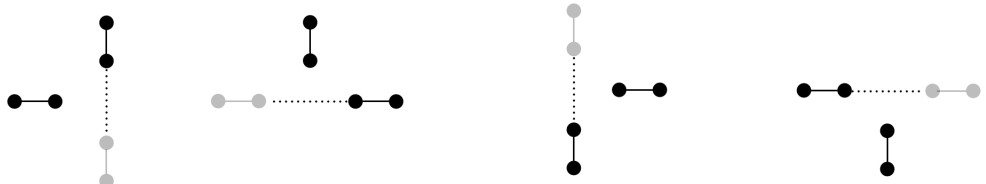

Figure 5: Brading process between dipoles. First we move $\mathfrak{p}^y$ upwards, then $\mathfrak{p}^x$ to the right. Finally $\mathfrak{p}^y$ moves downwards and $\mathfrak{p}^x$ returns to its original position.

where $\hat{x}$ and $\hat{y}$ are the coordinates of the relative distance from the initial $\mathfrak{q}$ and $\mathfrak{p}^x$ and $\mathfrak{p}^y$ particles mod $N$, similar to the position dependent braiding statistics present in Ref. [47]. Similarly, for the diagonal lineons $\theta(\mathfrak{q}, \mathfrak{d}^1) = \frac{2\pi}{N}(\hat{x} - \hat{y})$ and $\theta(\mathfrak{q}, \mathfrak{d}^2) = -\frac{2\pi}{N}(\hat{x} + \hat{y})$. These are explicit examples of position-dependent quantum numbers, as recently pointed out in [37].

## 3 Hierarchy of Time Scales and Effective Field Theories

In this section we study effective field theories for the microscopic model studied in the previous section. In general, given a lattice model with a lattice spacing $a$, the long-distance or continuum limit is reached by considering the limit of $a \to 0$. However, because of the mixing of scales UV/IR in the present lattice model, the effective field theories here cannot be completely defined without the presence of the lattice scale $a$. As we shall see, there are two qualitatively different field theories that describe two distinct regimes of time and both of them depend, explicitly or implicitly on $a$. The time scale that sets and separates these two regimes is the typical time that it takes for isolated excitations to hop from their original positions.

To see this, we consider the effect of local perturbations to the Hamiltonian (2),

$$H \to H + g_x \sum_{\vec{x}} X_{\vec{x}} + g_z \sum_{\vec{x}} Z_{\vec{x}} \,, \tag{28}$$

with $g_x \sim g_z$ small compared to the gap $J$. Such perturbations induce particles to hop. For the hopping of an isolated $\mathfrak{q}$-excitation, we need to go to higher orders in perturbation theory. Indeed, we need to consider a process that places one, two, three and so on up to $N$ Pauli operators at the first, second, third, and so on up to the $N$-th lattice site near the excitation. Thus, tunneling of an isolated $\mathfrak{q}$-particle will appear only at order $\sum_{n=1}^{N} n = N(N+1)/2 \sim N^2$ in perturbation theory. Correspondingly, it takes a time

$$\tau_{\text{monopole}} \sim \left(\frac{J}{g}\right)^{N^2} \,, \tag{29}$$

for the $\mathfrak{q}$-particle to move at zero temperature. Since $J \gg g$, this time scale increases quite rapidly with $N$ (the group order). In contrast, since the dipole can hop sequentially in the lattice, the characteristic time for a dipole to hop is

$$\tau_{\text{dipole}} \sim \left(\frac{J}{g}\right) \,. \tag{30}$$

Therefore, we see that isolated particles behave effectively as completely immobile excitations (fractons), taking a super-exponential time to hop. This means that even though this model is not an intrinsic fractonic system, for mild values of $N$ we would never see an isolated particle

hop. (We note that it is not uncommon for a system with slow dynamics to present emergent conservation laws. See, e.g., Refs. [48] and [49] for, respectively, theoretical and experimental settings where this phenomenon occurs). In the general classification, this model would be a two-dimensional type-I fracton system, where only composite particles are mobile. The dipole excitations do not take an exponential time to move and are identified as the mobile lineons.

In the following sections we explore the effective field theories for the lattice model in the two regimes $\tau \ll \tau_{\text{monopole}}$ and $\tau \gg \tau_{\text{monopole}}$. As we shall see, a conserving higher multipole momentum theory, which takes into account the immobility of excitations, depends explicitly on the lattice scale $a$ directly on the action. In contrast, in the regime where all excitations are able to hop, the effective field theory depends implicitly on the lattice scale $a$ through twisted boundary conditions on the fields.

### 3.1 Fractonic Regime: $\tau \ll \tau_{\text{monopole}}$

The effective field theory in this regime can be derived directly from the lattice. This is achieved by representing the microscopic $\mathbb{Z}_N$ degrees of freedom in terms of $U(1)$ fields in the continuum (Higgs mapping), according to

$$X_i = \exp(iaA_2) \quad \text{and} \quad Z_i = \exp(-iaA_1)\,, \tag{31}$$

where $a$ is the lattice spacing length and $A_{1,2}$ are dimension-one fields in mass units suitable for the continuum limit. This mapping is a faithful representation for describing the ground states of gapped phases. The requirement that the $\mathbb{Z}_N$-algebra is satisfied amounts to

$$[A_1(\vec{x}, t), A_2(\vec{y}, t)] = \frac{2\pi i}{N} \frac{\delta_{\vec{x}, \vec{y}}}{a^2}\,. \tag{32}$$

With this representation, it is straightforward to derive the effective field theory. First, we express the plaquette operator in terms of $U(1)$ fields. The leading term in an expansion in powers of $a$ is

$$B_{\vec{x}} \sim \exp ia^3 \left( \partial_1^2 A_2 - \partial_2^2 A_1 \right)\,. \tag{33}$$

With this, the Hamiltonian (2) becomes

$$H \sim -J \int d^2x \cos a^3 \left( \partial_2^2 A_1 - \partial_1^2 A_2 \right)\,. \tag{34}$$

As stated, the representation in (31) is faithful in describing the ground state of the system. We can construct an effective action describing the ground state of this Hamiltonian as

$$S_{CS} = -\frac{N}{2\pi} \int d^2x \, dt \left[ A_1 \partial_t A_2 + A_0 \left( D_1 A_2 - D_2 A_1 \right) \right]\,, \tag{35}$$

with $D_1 \equiv a\partial_1^2$ and $D_2 \equiv a\partial_2^2$. The first term in the action implies the equal-time commutation relation (32), whereas the second term is a constraint ensuring that we are in the ground state of the system, with $A_0$ being the corresponding Lagrange multiplier.

#### 3.1.1 Properties of the Effective Field Theory

The action (35) exhibits several interesting properties which we shall explore. Firstly, we have fixed the dimension of the Lagrange multiplier field $A_0$ to be same as the fields $A_1$ and $A_2$, namely, $[A_0] = [A_1] = [A_2] = 1$ in mass units. This implies that the lattice spacing $a$ appears explicitly in the effective field theory. Even though we are able to absorb the lattice spacing

into a redefinition of $A_0$ or into a rescaled time $t \to \frac{t}{a}$, it never disappears of the theory. To appreciate this point, we must consider not only the action, but also the gauge structure it implies. In fact, the action (35) is invariant under the following gauge transformations

$$A_0 \to A_0 + \partial_t \Lambda, \quad A_1 \to A_1 + D_1 \Lambda \quad \text{and} \quad A_2 \to A_2 + D_2 \Lambda, \tag{36}$$

up to boundary terms. Therefore, we see that even if we can scale out the lattice spacing of the action, it is still present in the gauge structure. This is a manifestation of the UV/IR mixing in the model, where the low-energy physics cannot be entirely defined without specifying UV information.

Similarly to the usual Chern-Simons theory, (35) is fully gapped and contains no local degrees of freedom, since the equations of motion lead to trivial configurations for the gauge-invariant electric and magnetic fields

$$\frac{N}{2\pi}B = \frac{N}{2\pi}E_1 = \frac{N}{2\pi}E_2 = 0, \tag{37}$$

with,

$$B \equiv D_1 A_2 - D_2 A_1, \quad E_1 \equiv D_0 A_1 - D_1 A_0, \quad \text{and} \quad E_2 \equiv D_0 A_2 - D_2 A_0. \tag{38}$$

As in the usual case, a finite gap is obtained by introducing "Maxwell" terms into the action, i.e., terms proportional to the square of electric and magnetic fields. In this sense, we consider the action

$$S = S_{CS} + \int dt\, d^2x \left[ \frac{1}{2g_E}\left(E_1^2 + E_2^2\right) + \frac{1}{2g_M}B^2 \right], \tag{39}$$

where $g_E$ and $g_M$ are dimension-one (in mass units) coupling constants. The gap can be determined from the poles of the propagator. In this gauge $A_0 = 0$, the equations of motion in momentum space read

$$\sum_{j=1}^{2}\left( \frac{1}{g_E}\delta_{ij}\omega^2 - \frac{1}{g_M}\left[ \sum_{k=1}^{2}P_k P_k \delta_{ij} - P_i P_j \right] + \frac{iN\omega}{2\pi}\epsilon_{ij} \right)A_j = 0, \tag{40}$$

with $P_j \equiv p_j^2$. This expression can be written in a matrix form as

$$\begin{pmatrix} \frac{\omega^2}{g_E} - \frac{p_y^4}{g_M} & \frac{p_x^2 p_y^2}{g_M} + \frac{iN\omega}{2\pi} \\ \frac{p_x^2 p_y^2}{g_m} - \frac{iN\omega}{2\pi} & \frac{\omega^2}{g_E} - \frac{p_x^4}{g_M} \end{pmatrix} \begin{pmatrix} A_1 \\ A_2 \end{pmatrix} = \begin{pmatrix} 0 \\ 0 \end{pmatrix}. \tag{41}$$

The propagator is essentially the inverse of the above matrix, which we denote by $G$. The poles follow from its determinant,

$$\det(G_{ij}) = \frac{\omega^4}{g_E^2} - \frac{\omega^2}{g_E g_M}\left( p_x^4 + p_y^4 \right) - \frac{N^2}{4\pi^2}\omega^2 = 0, \tag{42}$$

which implies that the dispersion is

$$\omega^2 = \frac{g_E}{g_M}\left( p_x^4 + p_y^4 \right) + \left( \frac{N g_E}{2\pi} \right)^2. \tag{43}$$

Therefore, the theory has a gap $\sim g_E$, which goes to infinity in the limit $g_E \to \infty$, where we recover the action (35).

### 3.1.2  Mobility Properties and Generalized Global Symmetries

As any local gauge-invariant quantity is trivial, we are led to study the global aspects and defects of the gauge theory (35). For this purpose, we define the theory on a 3-torus of sizes $L_t, L_x, L_y$. Then, we are allowed to consider large gauge transformations

$$\Lambda = 2\pi n_t \frac{t}{L_t} + 2\pi n_x \frac{x}{L_x} + 2\pi n_y \frac{y}{L_y}, \quad \text{with} \quad n_t, n_x, n_y \in \mathbb{Z}. \tag{44}$$

These transformations act nontrivially only on the temporal component $A_0$,

$$A_0 \to A_0 + \frac{2\pi n_t}{L_t}, \tag{45}$$

whereas $A_1$ and $A_2$ are unaffected. This implies that the defect

$$\exp\left(i \oint dt A_0(x,y)\right), \tag{46}$$

needs to be exponentiated to be invariant under all the gauge transformations, including the large ones above. On the contrary, the line operators

$$\oint dx A_1 \quad \text{and} \quad \oint dy A_2, \tag{47}$$

need only to be integrated over closed lines, but not exponentiated. At this point it is worth to emphasize that we are only exploring the gauge structure of the effective action to construct the extended gauge-invariant operators. We are not using the lattice memory of the maps (31), which imply compactness conditions for the fields, namely, $A_i \sim A_i + \frac{2\pi n}{a}$, with $n \in \mathbb{Z}$, so that only exponentials of extended operators would be allowed. Nevertheless, we shall consider the exponential of (47) in order to make connection with the lattice, but we will rely exclusively on the gauge structure of the effective action. We shall return to this point shortly.

Let us discuss first the properties of the defect (46). Gauge invariance dictates that its line cannot be deformed across the spatial directions. Thus, the defect (46) represents a probe excitation that is completely immobile. It is a fracton. By following [36], we can also understand this immobility from the perspective of a global symmetry, more precisely, a higher-form global symmetry.

The action is invariant under the global transformations

$$\begin{aligned} A_0 &\to A_0 + \frac{\lambda_0}{L_t} + \frac{2\pi n_x x}{L_t L_x} + \frac{2\pi n_y y}{L_t L_y} + \frac{2\pi m}{L_t} \frac{xy}{L_x L_y}, \\ A_i &\to A_i + \frac{\lambda_i}{L_i}, \end{aligned} \tag{48}$$

with $n_i, m \in \mathbb{Z}$, $\lambda_0 \in S^1$ and $\lambda_i \in \mathbb{R}$.

Notice that the transformations (48) are not gauge transformations, i.e., they cannot by annulled by any kind of gauge transformation. They really correspond to global symmetries. While their action on gauge-invariant local quantities is trivial, the defect (46) and the line operators (47) are charged under such symmetries. The defect transforms as

$$\exp\left(i \oint dt A_0\right) \to \exp i\left(\lambda_0 + \frac{2\pi n_x x}{L_x} + \frac{2\pi n_y y}{L_y} + \frac{2\pi m x y}{L_x L_y}\right) \exp\left(i \oint dt A_0\right). \tag{49}$$

In other words, the defect is charged under the global symmetry, with the charge depending of the positions $x$ and $y$ in both directions. This is precisely the unconventional property

that makes the excitation completely immobile. In fact, this implies that this defect cannot be moved in any direction to a different position without violating the global symmetry. The only possibility is through the displacements $x \rightarrow x + L_x$ or $y \rightarrow y + L_y$, but the points $x_i$ and $x_i + L_i$ correspond to the same spatial position in the torus. The exigence that the global charge is the same as we go around the two directions of the torus leads to the identifications

$$n_x \sim n_x + m \quad \text{and} \quad n_y \sim n_y + m, \tag{50}$$

which make both $n_x$ and $n_y$ to be defined mod $m$. This is equivalent to say that there are $m$ different charges in each point of space. Comparison with the lattice suggests that $m$ is identified with $N$. In the lattice, the single excitation carries a $\mathbb{Z}_N$ position-dependent charge.

Next we can study the properties of a dipole constructed from two defects disposed along the $x$-direction,

$$\exp\left( i \oint dt (A_0(t, x + x_0, y) - A_0(t, x, y)) \right), \tag{51}$$

where $x_0$ is the separation of the dipole. The charge of this defect configuration is

$$\exp i \left( \frac{2\pi n_x x_0}{L_x} + \frac{2\pi m x_0 y}{L_x L_y} \right), \tag{52}$$

which depends only on the position in the $y$-direction. So in this case the global symmetry prevents movement of this configuration in the $y$-direction, but it is free to move along the $x$-direction. Similar reasoning for a dipole disposed along the $y$-direction leads to the conclusion that it can move only in the $y$-direction. Therefore, dipole configurations in this system are allowed by the global symmetry to move only along the direction of their axis. This is precisely the movement of the dipole excitations of the lattice model.

We can make a closer connection between the dipole configuration in (51) and the lattice dipole operators (17) upon using the maps (31), which produce the exponentiated version of the operators in (47). By following [36], we can consider a more general gauge-invariant operator that needs not to be exponentiated

$$\oint_{\mathcal{C}} \left[ dt \left( \partial_x A_0 + \partial_y A_0 \right) + \frac{1}{a} dx A_1 + \frac{1}{a} dy A_2 \right], \tag{53}$$

where $\mathcal{C}$ is a closed curve in space-time. Now, consider a closed curve $\mathcal{C}_{t,x}$ lying in the $t$-$x$ plane, at fixed $y$, so that the above operator reduces to

$$\oint_{\mathcal{C}_{t,x}} \left[ dt \, \partial_x A_0 + \frac{1}{a} dx A_1 \right]. \tag{54}$$

In this case, we can construct the integrated operator

$$\oint_{\mathcal{C}_{t,x}} \left[ dt \left( A_0(t, x + x_0, y) - A_0(t, x, y) \right) + \frac{dx}{a} \int_x^{x+x_0} dx' A_1(t, x', y) \right]. \tag{55}$$

We see that the terms involving $A_0$ correspond precisely the structure appearing in the dipole configuration of (51), so that it is natural to consider the exponential of this operator,

$$\exp i \oint_{\mathcal{C}_{t,x}} \left[ dt \left( A_0(t, x + x_0, y) - A_0(t, x, y) \right) + \frac{dx}{a} \int_x^{x+x_0} dx' A_1(t, x', y) \right]. \tag{56}$$

The particular case where the line $\mathcal{C}_{t,x}$ is purely spatial reduces to the line operator coming from the lattice,

$$\exp i \oint \left[ \frac{dx}{a} \int_x^{x+x_0} dx' A_1(t, x', y) \right], \tag{57}$$

describing the mobility along the $x$-direction of a dipole oriented in this direction. The same reasoning for a line $\mathcal{C}_{t,y}$, lying in the $t$-$y$ plane, leads to similar conclusions for the dipoles oriented in the $y$-direction. Therefore, the higher-form global symmetries provide a precise way to understand the mobility of the dipoles in compliance with the lattice model.

Finally, we study quadrupole configurations. This type of defect can be constructed from four single defects disposed in the form of a quadrupole,

$$\exp \left( i \oint dt \left[ A_0(t, x+x_0, y+y_0) - A_0(t, x+x_0, y) - A(t, x, y+y_0) + A_0(t, x, y) \right] \right). \tag{58}$$

While it is charged under the global symmetries, its charge is independent of position,

$$\exp i \left( \frac{2\pi x_0 n_x}{L_x} + \frac{2\pi y_0 n_y}{L_y} \right), \tag{59}$$

in contrast with the previous defects considered. Therefore, the global symmetries do not impose any restriction on the mobility of this configuration, leading to the conclusion that quadrupoles can move freely.

### 3.1.3 Generalized Continuity Equation

In relativistic gauge theories, the study of line operators can be rephrased in terms of matter currents coupled to the gauge fields. In the present case, we can follow a similar reasoning and understand the immobility of excitations by coupling the gauge fields to external sources $(J_0, J_i)$ and studying their generalized continuity equation. This perspective is directly connected with the intuitive argument that fracton phenomenology follows from dipole conservation [50].

The coupling to an external current in the form,

$$S = S_{CS} + \int dt d^2x \left[ A_0 J_0 + A_1 J_1 + A_2 J_2 \right], \tag{60}$$

is gauge-invariant provided that the current satisfies a generalized version of the continuity equation,

$$\partial_t J_0 = D_1 J_1 + D_2 J_2. \tag{61}$$

This equation immediately leads to a global conserved charge $Q = \int d^2x J_0$. In addition, due to the form of the derivative operators $D_1$ and $D_2$, extra conserved charges emerge in the system, namely,

$$Q_f \equiv \int d^2x J_0 f(x, y), \quad \text{with} \quad f(x, y) = a x y + b x + c y, \tag{62}$$

for arbitrary $a, b, c \in \mathbb{R}$. These extra conserved charges correspond to higher multipole moments, responsible for constraining the mobility of excitations [50, 51]. However, due of the explicit dependence on coordinates, these charges may be ill-defined on compact manifolds or they may even be divergent in a infinite space [36]. In spite of the concerns about the precise meaning of these extra charges, they can be used, at least qualitatively, to understand the restriction on the mobility of the excitations discussed previously.

To this end, we consider the density $J_0$ corresponding to a single charge localized at $(\tilde{x}(t), \tilde{y}(t))$ in the instant of time $t$,

$$J_0(x, y, t) = \delta(x - \tilde{x}(t))\delta(y - \tilde{y}(t)). \tag{63}$$

The conservation of the charges $Q_f$ in (62) implies that

$$a\tilde{x}(t)\tilde{y}(t) + b\tilde{x}(t) + c\tilde{y}(t) = \text{constant}, \quad \forall\, a, b, c \in \mathbb{R}, \tag{64}$$

which can only be satisfied if

$$\tilde{x}(t) = \text{constant} \quad \text{and} \quad \tilde{y}(t) = \text{constant}, \tag{65}$$

since the parameters $a, b$ and $c$ are arbitrary and independent. This shows us that a single charge configuration compatible with the continuity equation is necessarily immobile.

Next we consider the density of a dipole configuration, with two opposite charged excitations located at $(\tilde{x}^1(t), \tilde{y}^1(t))$ and $(\tilde{x}^2(t), \tilde{y}^2(t))$. The density $J_0$ associated with this configuration is

$$J_0 = \delta(x - \tilde{x}^1(t))\delta(y - \tilde{y}^1(t)) - \delta(x - \tilde{x}^2(t))\delta(y - \tilde{y}^2(t)). \tag{66}$$

Once again, the conservations in (62) yield to the relation

$$a\left(\tilde{x}^1\tilde{y}^1 - \tilde{x}^2\tilde{y}^2\right) + b\left(\tilde{x}^1 - \tilde{x}^2\right) + c\left(\tilde{y}^1 - \tilde{y}^2\right) = \text{constant}, \tag{67}$$

where the time dependence is implicit. In contrast to the previous case, there are non-constant solutions for arbitrary $a, b$ and $c$, i.e.,

$$\tilde{y}^1(t) = \tilde{y}^2(t) = \text{constant} \quad \text{and} \quad \tilde{x}^1(t) - \tilde{x}^2(t) = \text{constant}, \tag{68}$$

or

$$\tilde{x}^1(t) = \tilde{x}^2(t) = \text{constant} \quad \text{and} \quad \tilde{y}^1(t) - \tilde{y}^2(t) = \text{constant}. \tag{69}$$

These solutions correspond to dipoles disposed along the $x$ and $y$-directions moving along their axis.

By following the same reasoning, we see that for a quadrupole configuration,

$$\begin{aligned}
J_0 &= \delta(x - \tilde{x}^1(t))\delta(y - \tilde{y}^1(t)) - \delta(x - \tilde{x}^2(t))\delta(y - \tilde{y}^1(t)) \\
&+ \delta(x - \tilde{x}^2(t))\delta(y - \tilde{y}^2(t)) - \delta(x - \tilde{x}^1(t))\delta(y - \tilde{y}^2(t)),
\end{aligned} \tag{70}$$

there are no restrictions on the mobility since the conservation of (62) implies

$$\left(\tilde{x}^1(t) - \tilde{x}^2(t)\right)\left(\tilde{y}^1(t) - \tilde{y}^2(t)\right) = \text{constant}. \tag{71}$$

As long as the quadrupole configuration is preserved, it can move freely. Therefore, we see that the study of the generalized conservation laws (62) arising from the generalized continuity equation (61) leads precisely to the same conclusions concerning the mobility of excitations that we have obtained through the analysis of the higher-form global symmetries in the previous section.

## 3.2 Limit of Mobile Excitations: $\tau \gg \tau_{\text{monopole}}$

We finally study the limit where all excitations are completely mobile $\tau \gg \tau_{\text{monopole}}$, since all the restrictions on the mobility of excitations $\mathfrak{q}$, $\mathfrak{p}$ and $\mathfrak{d}$ vanish. We thus expect a continuum theory describing excitations that are completely mobile and that share nontrivial mutual statistics. Among all the lineons $\mathfrak{p}^x, \mathfrak{p}^y, \mathfrak{d}^1$ or $\mathfrak{d}^2$, only a single pair is independent, given that the other lineons can be obtained from fusion, as discussed previously. For our purposes, it is convenient to choose the pair $\mathfrak{p}^x$ and $\mathfrak{p}^y$. Now, among the particles $\mathfrak{q}, \mathfrak{m}, \mathfrak{p}^x, \mathfrak{p}^y$ it follows that the non-vanishing mutual statistics are $\theta(\mathfrak{q}, \mathfrak{m}) = 2\pi/N$ and $\theta(\mathfrak{p}^x, \mathfrak{p}^y) = 2\pi/N$.

The basic idea to construct the effective theory in this regime is to associate a gauge field to each one of the excitations, while keeping track the information about their mutual statistics. These features can be naturally embodied into the $K$-matrix formulation of topological fluids, whose effective continuum action is given in terms of a collection of Chern-Simons gauge fields $a^{(a)}$, coupled through a $K$-matrix governing the mutual statistics [52]. Let us associate the first two fields $a^{(1)}$ and $a^{(2)}$ to the $\mathfrak{q}$ and $\mathfrak{m}$-particles and the fields $a^{(3)}$ and $a^{(4)}$ to the $\mathfrak{p}^x$ and $\mathfrak{d}^1$-particles, respectively. Then we write the $K$-matrix Chern-Simons action

$$S = \int d^3x \frac{K_{ab}}{4\pi} \epsilon^{\mu\nu\alpha} a_\mu^{(a)} \partial_\nu a_\alpha^{(b)}, \tag{72}$$

where the $4 \times 4$ symmetric $K$-matrix

$$K = \begin{pmatrix} N\sigma^x & 0_{2\times2} \\ 0_{2\times2} & N\sigma^x \end{pmatrix}, \tag{73}$$

encodes the mutual statistics among the $a$-th and $b$-th particles according to $\theta_{ab} = 2\pi \left(K^{-1}\right)_{ab}$. Notice that the action (72) makes no reference to the lattice spacing, in contrast with the effective action in the fractonic regime. However, to properly incorporate the peculiar topological properties of the anyons present on the microscopic theory, some memory of the lattice spacing is needed. Namely, for a particular anyon to be able to annihilate with its anti-particle, they must be dragged around the system multiple times. In this regime, this kind of information can be incorporated through nontrivial boundary conditions on the fields present in the effective action (72).

### 3.2.1 Ground State Degeneracy

In addition to the mutual statistics, the effective action (72) can be used to compute the ground state degeneracy. The translation group is realized non-linearly in the effective theory and it plays an important role in determining the ground state degeneracy. If all the four fields $a^{(a)}$ satisfy periodic boundary conditions, then the ground state degeneracy is simply $\det K = N^4$. This always happens when, after a translation around the system by $aL_x$ (and similarly for the $y$-direction), all the excitations are mapped into themselves ($L_x = pN$, with $p \in \mathbb{Z}$), which agrees with the result in (12).

In the limit $\tau \gg \tau_{\text{monopole}}$, since all particles are completely mobile, the excitations can always be moved around the system and go back to the original point. However, it is not guaranteed that after returning to their original position these particles belong to the same superselection sectors, so that they may not be able to annihilate themselves back to the vacuum. For boundary conditions where the anyons are not mapped into themselves ($L_x \neq pN$), the field species are mixed under translation and a translation by $aL_x$ no longer acts as an identity on the field space.

Following the discussion in [37], we can derive the ground state degeneracy in the continuum theory by studying the boundary conditions of the gauge fields. Since the translation

operations do not act linearly on the gauge fields, it is difficult to write their general transformation under arbitrary shifts. We know, nevertheless, under which translations the anyons are invariant, a property that must be also obeyed by the corresponding gauge fields. Respecting the translation properties of the anyons, the fields must obey the following periodic conditions

$$
\begin{aligned}
a^{(1)}(x,y) &= a^{(1)}(x + a\,\mathrm{lcm}(N, L_x), y) = a^{(1)}(x, y + a\,\mathrm{lcm}(N, L_y)), \\
a^{(2)}(x,y) &= a^{(2)}(x + a\,L_x, y) = a^{(2)}(x, y + a\,L_y), \\
a^{(3)}(x,y) &= a^{(3)}(x + a\,L_x, y) = a^{(3)}(x, y + a\,\mathrm{lcm}(N, L_y)).
\end{aligned}
\tag{74}
$$

The transformations for $a^{(1)}$ in both directions follow from the fact that the $\mathfrak{q}$-particle must hop $\mathrm{lcm}(N, L))/L$ times around the system in order to go back to its original position and annihilate itself, according to (16). The transformation for $a^{(2)}$ is the usual periodic boundary condition satisfied by the $\mathfrak{m}$-particle on the torus (21). The transformations for $a^{(3)}$ follow from (18) for the $\mathfrak{p}^x$-particles. It is tempting to propose similar transformations for the fourth field $a^{(4)}$, following the translations of the $\mathfrak{p}^y$-particle in (18)

$$
\begin{aligned}
a^{(4)}(x,y) &= a^{(4)}(x, y + a\,L_y), \tag{75} \\
a^{(4)}(x,y) &= a^{(4)}(x + a\,\mathrm{lcm}(N, L_x), y). \tag{76}
\end{aligned}
$$

Although the first equality above holds, the naive expression (76) is not restrictive enough, as it is possible to construct operators that hop particles along the horizontal direction with step sizes possibly smaller than $N$. The point is that in (76) we are leaving out the information that the lattice diagonal operators $S(\Gamma)$ introduce a new hopping step. To correctly incorporate the periodicity in the $x$-direction for $a^{(4)}$, let us note that in order to move $\mathfrak{p}^y$ in the horizontal direction, we can use that $\mathfrak{p}^y = \overline{\mathfrak{p}^x} \times \mathfrak{d}^1$ and move $\overline{\mathfrak{p}^x}$ and $\mathfrak{d}^1$ independently instead. As we noticed before, $\overline{\mathfrak{p}^x}$ can move freely along the $x$-direction and thus will provide us no constraint on the periodicity of $a^{(4)}$. It is the interplay among the $U(\lambda)$ and $S(\Gamma)$ operators in the hopping of $\mathfrak{d}^1$ that gives us the proper result.

To appreciate this point, we first note that the existence of $U(\lambda_y)$ allows the motion of $\mathfrak{d}^1$ along the vertical direction in step sizes of $N$ and, consequently, separates the vertical direction into $N L_y / \mathrm{lcm}(N, L_y) = \gcd(N, L_y)$ disconnected sub-lattices. Effectively, it introduces a new step size $\tilde{N} \equiv \gcd(N, L_y)$ operator that allows the particles to hop along $y$-direction. Next, since $\mathfrak{d}^1$ moves diagonally, this $y$-direction $\tilde{N}$-step operator can be used, effectively, to hop $\mathfrak{d}^1$-particles in steps of size $\tilde{N}$ along the $x$-direction too. Thus, with this new scale $\tilde{N}$, the $\mathfrak{d}^1$-particles, as well as $\mathfrak{p}^y$, must obey the lattice translation

$$
\hat{T} : (\hat{x}, \hat{y}) \mapsto (\hat{x} + \mathrm{lcm}(L_x, \tilde{N}), \hat{y}),
\tag{77}
$$

in order to return to their original position after repeated $\tilde{N}$-sized steps along the horizontal direction. This implies that, in the continuum, the corresponding $a^{(4)}$-field has a smaller periodicity

$$
a^{(4)}(x,y) = a^{(4)}(x + a\,\mathrm{lcm}(\tilde{N}, L_x), y),
\tag{78}
$$

when compared with the naive boundary condition (76).

The ground state degeneracy is dictated by the topological configurations involving only the zero mode of the fields, namely, solutions of the equations of motion depending only on the time $a^{(a)}(x, y, t) \to a^{(a)}(t)$. We then parameterize the fields taking into account the periodicity

of the holonomies according to (74), (75) and (78)

$$
\begin{aligned}
a_x^{(1)} &= \frac{\bar{a}_x^{(1)}(t)}{a\,\mathrm{lcm}(N,L_x)}, & a_y^{(1)} &= \frac{\bar{a}_y^{(1)}(t)}{a\,\mathrm{lcm}(N,L_y)}, \\
a_x^{(2)} &= \frac{\bar{a}_x^{(2)}(t)}{a\,L_x}, & a_y^{(2)} &= \frac{\bar{a}_y^{(2)}(t)}{a\,L_y}, \\
a_x^{(3)} &= \frac{\bar{a}_x^{(3)}(t)}{a\,L_x}, & a_y^{(3)} &= \frac{\bar{a}_y^{(3)}(t)}{a\,\mathrm{lcm}(N,L_y)}, \\
a_x^{(4)} &= \frac{\bar{a}_x^{(4)}(t)}{a\,\mathrm{lcm}(L_x,\gcd(L_y,N))}, & a_y^{(4)} &= \frac{\bar{a}_y^{(4)}(t)}{a\,L_y}.
\end{aligned}
\tag{79}
$$

The requirement that the holonomies must go around the system multiple times before they close is a memory of how the $N$-hopping strings could cover the $L_x \times L_y$ lattice. Replacing these topological solutions back into the action (72), we get a simple quantum mechanical system

$$
\begin{aligned}
S = \frac{1}{2\pi}\int \; dt \; \Big[ &\gcd(N,L_y)\bar{a}_y^{(1)}\dot{\bar{a}}_x^{(2)} - \gcd(N,L_x)\bar{a}_x^{(1)}\dot{\bar{a}}_y^{(2)} \\
+ \; &\gcd(N,L_x,L_y)\bar{a}_y^{(3)}\dot{\bar{a}}_x^{(4)} - N\bar{a}_x^{(3)}\dot{\bar{a}}_y^{(4)} \Big],
\end{aligned}
\tag{80}
$$

where we have used the identity $\gcd(a,b)\mathrm{lcm}(a,b) = ab$, for integers $a$ and $b$. Upon canonical quantization, we get the algebra among the gauge-invariant operators

$$
\begin{aligned}
\exp\!\left(i\bar{a}_x^{(1)}\right)\exp\!\left(i\bar{a}_y^{(2)}\right) &= \exp\!\left(-\frac{2\pi i}{\gcd(N,L_x)}\right)\exp\!\left(i\bar{a}_y^{(2)}\right)\exp\!\left(i\bar{a}_x^{(1)}\right), \\
\exp\!\left(i\bar{a}_y^{(1)}\right)\exp\!\left(i\bar{a}_x^{(2)}\right) &= \exp\!\left(\frac{2\pi i}{\gcd(N,L_y)}\right)\exp\!\left(i\bar{a}_x^{(2)}\right)\exp\!\left(i\bar{a}_y^{(1)}\right), \\
\exp\!\left(i\bar{a}_x^{(3)}\right)\exp\!\left(i\bar{a}_y^{(4)}\right) &= \exp\!\left(-\frac{2\pi i}{N}\right)\exp\!\left(i\bar{a}_y^{(4)}\right)\exp\!\left(i\bar{a}_x^{(3)}\right), \\
\exp\!\left(i\bar{a}_y^{(3)}\right)\exp\!\left(i\bar{a}_x^{(4)}\right) &= \exp\!\left(\frac{2\pi i}{\gcd(N,L_x,L_y)}\right)\exp\!\left(i\bar{a}_x^{(2)}\right)\exp\!\left(i\bar{a}_y^{(1)}\right).
\end{aligned}
\tag{81}
$$

The ground state degeneracy is the product of the representation sizes of each one of the algebras

$$
\dim \mathcal{H}_0 = N\,\gcd(N,L_x)\,\gcd(N,L_y)\,\gcd(N,L_x,L_y),
\tag{82}
$$

which precisely matches with the ground state space dimension of the lattice model (12). The parameters $L_x$ and $L_y$ correspond to UV information of the underlying regularization, as they are dimensionless parameters that count how many sites the underlying lattice contains. The proper dimensionful physical length of the system are related to the dimensionless parameters $L_x$ and $L_y$ through the lattice spacing as $aL_x$ and $aL_y$. Although the continuum theory (72) does not depend explicitly on the lattice spacing $a$, it depends implicitly on it since (82) involves the ratio between the physical length of the system and the lattice spacing $a$. As we saw directly from the microscopic model, it does not come as a surprise that the low-energy physics depends on the UV information through regularization details.

# 4 Conclusions

In this paper we proposed an exactly solvable two-dimensional model on the lattice involving $\mathbb{Z}_N$ degrees of freedom. The model is interesting in that it exhibits topological order, at the same time that its low-energy physics is quite sensitive to the details of the lattice (UV information). Besides nontrivial statistics among the emergent quasiparticles, they also present restricted mobility, resembling the fractonic physics in higher dimensional models. Although this model is not an intrinsic fracton system, since there are string operators that are able to move isolated particles, it behaves effectively as a fracton in a certain regime dictated by the periodicity $N$ of the group ($\mathbb{Z}_N$). Indeed, fixing a value for $N$, it defines a typical time $\tau_{\text{monopole}}$ for isolated particles to move that divides the observed long-distance physics into two distinct regimes.

In the regime of time scales $\tau \ll \tau_{\text{monopole}}$, the system behaves effectively as a type-I fracton. At zero temperature, it would take an exponentially large time for an isolated particle to hop from its original position. In this regime, the effective field theory is a generalization of a Chern-Simons theory incorporating the fracton physics by means of the presence of higher spatial derivative operators, which lead to the existence of higher-form global symmetries. These global symmetries, in turn, yield restrictions on the mobility of the excitations, since the corresponding defect operators are charged under such symmetries, with position-dependent charges. For example, for an isolated excitation, there is no possible motion compatible with the global symmetry. Dipoles, on the other, can move only along certain lines, whereas quadrupoles can move freely.

In time scales where all the particles become mobile, $\tau \gg \tau_{\text{monopole}}$, we use the nontrivial mutual statistics among the excitations of the system to construct a mutual Chern-Simons effective field theory. Although the mobility restrictions of the excitations vanish in this regime, the system still embodies their exotic properties through the nonlinear implementation of the translation group, which is translated to the Chern-Simons theory in terms of nontrivial periodic boundary conditions for the gauge fields. Taking this into account, we were able to compute the ground state degeneracy, recovering the lattice result.

Although some works corroborate the inexistence of intrinsic topologically ordered fractonic systems in two-dimensions [22, 53], the model studied in this work provides a different perspective on this problem. We have conceived the more modest possibility of the effective realization of the fractonic behavior in $d = 2$, where in practice a quasi-excitation takes an exponential time to move out from its position.

## Acknowledgments

G.D. is grateful to Julio Toledo, Alexey Khudorozhkov, Hongji Yu, Kai-Hsin Wu, and Salvatore D. Pace for all the insights and helpful discussions. This work is supported by the DOE Grant No. DE-FG02-06ER46316 (G. D. and C.C.). P. G. is partially supported by the CNPq. W.B.F. is supported by FUNPEC foundation under grant 182022/1707.

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
