# Peer review of "Effective Fractonic Behavior in a Two-Dimensional Exactly Solvable Spin Liquid"

_SciPost Physics, doi:SciPost Phys. 14, 002 (2023)_

## Round 2 · Referee Report · Anonymous (Referee 1) · 2022-8-18

Report

The authors present a systematic analysis of a 2+1D lattice model that arises as a projection of the 3+1D Chamon code onto the plane. Consequently, the model displays features that are qualitatively similar to those of fracton models. The authors also identify two distinct regimes that are separated by an exponentially long-time scale and are governed by different effective actions. The results are presented clearly and the paper is generally well-written.

This is an interesting paper that should be published once the following (mostly minor) points are addressed:

  1. While the model itself is new, its physics is that of a symmetry-enriched topological (SET) phase where lattice translations permute inequivalent anyon types. Although the authors do not make this identification explicitly, viewed thusly the position-dependent braiding statistics and lattice-size dependent ground state degeneracy are not surprising. I would suggest that the authors make reference to the existing literature and situate their model in this broader context.

  2. The authors do not state what topological order the model possesses; it is clearly $\mathbb{Z}_N^2$, but this should be mentioned somewhere.

  3. Similarly, how many distinct anyon types are present in the model? There should be $N^4$, which would be consistent with the field theory, but the authors should directly verify this in the lattice model.

  4. While the action of translation symmetry on the anyons is discussed, how do the two mirror symmetries $M_x$ and $M_y$ act? Are there any other lattice symmetries that act non-trivially on the anyons?

  5. With regards to the effective fractonic Chern-Simons theory, the authors should cite the following two papers where a similar action was first written down in terms of symmetric tensor gauge fields: Phys. Rev. B 96, 125151 (2017) and Phys. Rev. B 97, 085116 (2018)

  6. The authors should cite SciPost Phys. 10, 011 (2021) and mention Ref.[22] in the conclusions when discussing the existence of intrinsically fracton phases in 2+1D (under mild assumptions, both offer proofs that this is not possible).

  7. Finally, the idea of realizing "effective" fractonic behavior for exponentially long-time scales has previously been discussed theoretically and even demonstrated experimentally both in one and two dimensions (although the mechanism differs from that proposed by the authors). See e.g., Phys. Rev. X 10, 011042 (2020).

  • validity: high
  • significance: good
  • originality: good
  • clarity: high
  • formatting: excellent
  • grammar: excellent

Author:  Guilherme Delfino  on 2022-10-06  [id 2887]

(in reply to Report 1 on 2022-08-18)
Category:
answer to question
pointer to related literature

We thank the referee for the questions, suggestions, and references. We addressed these points below.

1. While the model itself is new, its physics is that of a symmetry-enriched topological (SET) phase where lattice translations permute inequivalent anyon types. Although the authors do not make this identification explicitly, viewed thusly the position-dependent braiding statistics and lattice-size dependent ground state degeneracy are not surprising. I would suggest that the authors make reference to the existing literature and situate their model in this broader context.

Thank you for the suggestion. We added a brief discussion about this point in the introduction of the paper.

2. The authors do not state what topological order the model possesses; it is clearly $ \mathbb{Z}_N^2$ but this should be mentioned somewhere.

We now state what topological order the model possesses explicitly in the end of section II.B.

3. Similarly, how many distinct anyon types are present in the model? There should be $N^4$, which would be consistent with the field theory, but the authors should directly verify this in the lattice model.

In the lattice model, there are as many distinct anyons as degenerate ground states $ \dim \mathcal{H}_0 $. For the case in which $ L_x=L_y $ are multiple of $N$, this reduces to the usual $ N^4 $ result. For the general case, however, a more complicated relation for the number of anyons arises according to Eq. (12), a number bounded by $ N $ and $ N^4 $. Although the topological order is $ \mathbb{Z}_N\times \mathbb{Z}_N $, the number of independent quasi-particles is reduced because of the existence of the $ N $-hopping line operators.

**4. While the action of translation symmetry on the anyons is discussed, how do the two mirror symmetries $M_x$ and $M_y$ act? Are there any other lattice symmetries that act non-trivially on the anyons? **

In the paper we have only considered the translation operations that act as identity on the anyon space, as they are relevant in the analysis of topological properties, as GSD and statistical angles. We believe that an investigation of the lattice symmetries action on anyons would require an approach similar to that in arXiv:2204.07111, where the authors introduce anyons that are explicitly dependent on position. Addressing the second question, the answer is positive: there are discrete lattice symmetries that act non-trivially on the anyons, as rotations and mirror symmetries e.g. $ C_4:\mathfrak{p}_x\rightarrow \mathfrak{p}_y $ and $ M_x:\mathfrak{p}_x\rightarrow \overline{\mathfrak{p}}_x$. In the paper, however, we did not consider such analysis as it is independent of the topological properties of the system, which is the main focus of the paper.

**5. With regards to the effective fractonic Chern-Simons theory, the authors should cite the following two papers where a similar action was first written down in terms of symmetric tensor gauge fields: Phys. Rev. B 96, 125151 (2017) and Phys. Rev. B 97, 085116 (2018) **

Thank you for the references. We have included them in the revised version of the manuscript.

** 6. The authors should cite SciPost Phys. 10, 011 (2021) and mention Ref.[22] in the conclusions when discussing the existence of intrinsically fracton phases in 2+1D (under mild assumptions, both offer proofs that this is not possible).**

Thank you for the reference. We have included the references in our discussion in the conclusions.

7. Finally, the idea of realizing "effective" fractonic behavior for exponentially long-time scales has previously been discussed theoretically and even demonstrated experimentally both in one and two dimensions (although the mechanism differs from that proposed by the authors). See e.g., Phys. Rev. X 10, 011042 (2020).

Thank you for calling our attention to this point. We included the suggested reference, as well as a reference to ArXiv:2009.05577.

---

## Round 2 · Referee Report · Anonymous (Referee 2) · 2022-8-19

Strengths

1-solid, 2-comprehensive

Report

The authors gave a solid and comprehensive study on a model whose excitations have restricted mobility. In the exact solvable limit, there are subextensive ground state degeneracy which is a function of system size. There are excitations only allowed to move along certain directions. Then, the authors formulated an effective field theory to describe the fractonic phase and also the phase where all the excitations turn into mobile. Although I think the paper meets the standard of SciPost, I am still confused about a few points. In order to better understand the results, I list my questions in the following and hope the author can make further clarification.

  1. Can the lattice model in Eq.2 be obtained by Higgsing a U(1) model/theory?
  2. In the effective theory, there is an immobile defect (Eq.47). It looks like a Wilson loop along time direction. Is it an instanton? Can I understand it in the language of the lattice model?
  3. As the defect carries the charge of a global higher form symmetry, is this symmetry an exact microscopic (UV) symmetry of the lattice model? If so, what is the symmetry transformation in terms of lattice operators? How can I see it leaves the lattice Hamiltonian invariant?
  4. By tuning the perturbation in Eq.29 very large, all the excitations gets full mobile. Can we understand this phase transition by a picture of the anyon condensation?
  • validity: good
  • significance: ok
  • originality: good
  • clarity: -
  • formatting: -
  • grammar: -

Author:  Guilherme Delfino  on 2022-10-06  [id 2888]

(in reply to Report 2 on 2022-08-19)
Category:
question

We thank the referee for the questions. We addressed the points below.

** 1. Can the lattice model in Eq.2 be obtained by Higgsing a U(1) model/theory? **

This is an interesting question. The authors of ArXiv: 2110.02658 and 2204.07111 studied lattice Hamiltonians that were derived from Higgsing a rank 2 $ U(1) $ tensor gauge theory on a lattice. We believe that it would be possible to extend our model to a $ U(1) $ version and Higgs it down to a $ \mathbb{Z}_N $ phase, similarly to their construction. In our paper, however, we did not explore this direction since we already had the $ \mathbb{Z}_N $ exactly solvable lattice model in Eq. (2).

** 2. In the effective theory, there is an immobile defect (Eq.47). It looks like a Wilson loop along time direction. Is it an instanton? Can I understand it in the language of the lattice model?**

The defect in Eq (47) is, indeed, a Wilson loop along the time direction. It emerges as a non-local gauge-invariant quantity in the Chern-Simons-like effective field theory. In principle, we do not see a connection of this defect with an instanton. The connection of such defect with the lattice excitations is not immediate, since the lattice model is defined on a Hilbert space (a constant time-slice of the field theory). Also, as opposed to the lattice model excitations, the external probe is not in the “intrinsic” spectrum of the field theory. We can understand the connection between the defect and excitations in the lattice model by noticing that the defect in Eq. (47) corresponds precisely to the worldline of an isolated particle at rest at position $\vec{x}_0$. One can see this by adding an external current to the gauge theory in Eq. (36), as in Eq. (61). For a particle at rest, $ J_0 (\vec{x})= \delta^2(\vec x-\vec{x}_0) $, and the added gauge-invariant coupling in the partition function reads

$$ \exp\left (i \int d^2x dt \, J_0 A_0 \right )=\exp\left (i \int d^2x dt \, \delta^2(\vec x - \vec{x}_0)A_0 (\vec{x}) \right )= \exp\left (i \oint dt A_0 (\vec{x}_0) \right )$$
which is precisely the defect in Eq. (47), i.e., a Wilson loop along the time direction. In the lattice, this corresponds to a $\mathfrak{q}$-particle at rest. In the continuum, however, since we are now in a scale of time $ \tau\ll \tau_{\text{monopole}} $, these $ \mathfrak{q} $-particles effectively do not move. In the paragraph before equation (49) we present a brief discussion on this point. In constrast, for a dipole configuration, which is composed of two $ \mathfrak{q} $-particles, the connection of the defect with the lattice operators is more direct, as discussed in Eq. (54)-(57).

3. As the defect carries the charge of a global higher form symmetry, is this symmetry an exact microscopic (UV) symmetry of the lattice model? If so, what is the symmetry transformation in terms of lattice operators? How can I see it leaves the lattice Hamiltonian invariant?

The transformations for the spatial components of the gauge field $A_i$ in Eq (49) can be understood in terms of lattice operators by keeping in mind their relation in Eq. (32). They consist of

$$ X\rightarrow U_1XU_1^{\dagger} = e^{i\alpha_x}X, \quad Z\rightarrow U_2ZU_2^{\dagger} = e^{-i\alpha_z}Z,$$
where the unitary operators are given by
$$ U_1 = Z^{\frac{N\alpha_x}{2\pi}}\quad \text{and} \quad U_2= X^{\frac{N\alpha_z}{2\pi}}.$$
(These also imply shift symmetries for the time component $A_0$, as the derivatives responsible for these symmetries in the field theory can also apply on $A_0$ if we integrate by parts.) We can also see the dipole symmetry in the lattice from the transformations $ X\rightarrow U_1^{(1)} X U_1^{(1)\dagger} = e^{i\beta_x \hat{x}}X $ and $ Z \rightarrow U_2^{(1)} Z U_2^{(1) \dagger} = e^{-i\beta_z \hat{y}}Z $, which are related to the continuum dipole symmetries in Eq. (49), and are implemented by
$$ U_1^{(1)} = Z^{\frac{N\beta_x \hat{x}}{2\pi}}\quad \text{and} \quad U_2^{(1)} = X^{\frac{N\beta_z \hat{y}}{2\pi}}.$$
The off-diagonal quadrupole symmetry can be similarly understood, $X\rightarrow U_1^{(2)} X U_1^{(2) \dagger} = e^{i\gamma_x \, \hat x\hat y}X$ and $Z\rightarrow U_2^{(2)} Z U_2^{(2) \dagger} = e^{i\gamma_z \, \hat x\hat y}Z$. All these $ U $ operators are symmetries of the lattice Hamiltonian, $[H, U]=0$.

4. By tuning the perturbation in Eq. (29) very large, all the excitations gets full mobile. Can we understand this phase transition by a picture of the anyon condensation?

Indeed, in the model in Eq. (29) there is a phase transition at a critical value $g_c$, beyond which anyons condense and the system is no longer in the $ \mathbb Z_N\times \mathbb Z_N$ topologically ordered state. For large $ g\gg g_c $, the phase corresponds to a paramagnet where no restriction on the mobility of excitations is imposed. One question is whether the monopoles become fully mobile before this transition at $ g_c $. The time $ \tau_{\text{monopole}} $ becomes of the order of $ \tau_{\text{dipole}} $ if $ g\approx J $. However we suspect that $g_c$ is smaller than $J$ and this situation never arises. More specifically, the topological order present in the ground state of the system for $ g< g_c $ is $\mathbb{Z}_N\times \mathbb Z_N$, enriched with translational symmetry. We note the ground state $ | GS \rangle $ of the perturbed Hamiltonian Eq (29), in the limit of very large $g_x$, becomes a condensate of $\mathfrak{q}$ and $ \mathfrak{m}$-anyons that are created along the $ y $ direction and also a condensate of $ \mathfrak{p}_y $-particles. More precisely, these excitations are created by the operators $ \mathcal{A}_x={U(\lambda_y), V(\gamma_y), W_y} $, that are all products of Pauli $X$ operators, showing us that the ground state is a condensate of particles

$$ \langle GS| \mathcal{A}_x| GS \rangle =1.$$
Similarly, in the limit of large $g_z$, the ground state $ | \overline{GS} \rangle $ is a condensate of $\mathfrak{q}$ and $ \mathfrak{m}$-anyons created along the $ x $ direction as well as $ \mathfrak{p}_x $-anyons
$$\langle \overline{GS}| \mathcal{A}_z| \overline{GS} \rangle =1,$$
for $ \mathcal{A}_z={U(\lambda_x), V(\gamma_x), W_x} $. In both these regimes, the topological order is destroyed $ \mathbb Z_N\times \mathbb Z_N\rightarrow \text{Trivial}$. We note that these arguments cannot rule out the existence of intermediate phases between the $ \mathbb Z_N\times \mathbb Z_N $ and the trivial phase.

---

## Round 3 · List of Changes

We added a brief discussion in the Introduction, associating our model to an example of a SET phase;
We added a discussion on the Z_N topological order present in the low-energy states of the model in Section II.B;
We also expanded on how some of the lattice symmetries act on the anyons in Section II.C;
Finally, we included further references.

---

## Editorial Decision

published